# Enhanced Sensitivity and Accuracy of Tb^3+^-Functionalized Zirconium-Based Bimetallic MOF for Visual Detection of Malachite Green in Fish

**DOI:** 10.3390/foods13172855

**Published:** 2024-09-09

**Authors:** Yue Zhou, Yuanyuan Jiang, Xiangyu Chen, Hongchen Long, Mao Zhang, Zili Tang, Yufang He, Lei Zhang, Tao Le

**Affiliations:** Chongqing Key Laboratory of Conservation and Utilization of Freshwater Fishes, Animal Biology Key Laboratory of Chongqing Education Commission of China, College of Life Sciences, Chongqing Normal University, Chongqing 401331, China; 2022210513076@stu.cqnu.edu.cn (Y.Z.); 2022210513056@stu.cqnu.edu.cn (Y.J.); 2022210513051@stu.cqnu.edu.cn (X.C.); 2022210513063@stu.cqnu.edu.cn (H.L.); 2023110513037@stu.cqnu.edu.cn (M.Z.); 2023210513073@stu.cqnu.edu.cn (Z.T.); 2023210513061@stu.cqnu.edu.cn (Y.H.); letao@cqnu.edu.cn (T.L.)

**Keywords:** ratiometric fluorescence, UiO-OH@Tb, FRET, smartphone

## Abstract

The ratiometric fluorescent probe UiO-OH@Tb, a zirconium-based MOF functionalized with Tb^3+^, was synthesized using a hydrothermal method. This probe employs the fluorescence resonance energy transfer (FRET) mechanism between Tb^3+^ and malachite green (MG) for the double-inverse signal ratiometric fluorescence detection of MG. The probe’s color shifts from lime green to blue with an increasing concentration of MG. In contrast, the monometallic MOFs’ (UiO-OH) probe shows only blue fluorescence quenching due to the inner filter effect (IFE) after interacting with MG. Additionally, the composite fluorescent probe (UiO-OH@Tb) exhibits superior sensitivity, with a detection limit (LOD) of 0.19 μM, which is significantly lower than that of the monometallic MOFs (25 μM). Moreover, the content of MG can be detected on-site (LOD = 0.94 μM) using the RGB function of smartphones. Hence, the UiO-OH@Tb probe is proven to be an ideal material for MG detection, demonstrating significant practical value in real-world applications.

## 1. Introduction

Malachite green (MG), a synthetic N-methyldiaminotriphenylmethane analog, was previously utilized extensively in aquaculture for its efficacy in combating water mold, melioidosis, and gill mold in fish [1]. However, studies have shown that MG is extremely toxic and may pose serious carcinogenic and teratogenic risks to animal and human health [2]. Despite explicit bans on MG use in aquaculture in many countries, illegal application persists due to the lack of cost-effective alternatives [3]. Therefore, the development of precise and efficient methods for monitoring MG residues is essential for safeguarding public health.

Various techniques have been established for MG detection, including high-performance liquid chromatography (HPLC) [4], surface-enhanced Raman spectroscopy (SERS) [5], electrochemical methods [6], and fluorescence analysis [7]. Among these methods, fluorescence analysis has gained widespread recognition and favor from researchers and industry experts due to its rapidity, high sensitivity, and efficiency [8].

Conventional fluorescence sensors rely on the fluorescence “on-off” or “off-on” mechanism, which is intuitive and convenient but suffers from poor anti-interference capability. These sensors are susceptible to concentration, environment, and other external factors, thereby limiting their application in sensing fields to a certain extent [9]. To address these limitations, ratiometric fluorescence sensors have emerged. Chi Jie et al. constructed a ratiometric fluorescent probe (UiO-66-NH_2_@RhB) with fluorescence/colorimetric bimodal detection by embedding rhodamine B (RhB) into UiO-66-NH_2_, achieving a limit of detection (LOD) of 4.87 μM for arginine, which is significantly lower than the LOD of UiO-66-NH_2_ alone for arginine (40.61 μM) [10]. Ameen and colleagues developed a novel UoZ-2 probe for the highly sensitive visual ratiometric detection of tetracycline (TC) with a LOD of 0.014 μM, surpassing Zn-MOF (0.028 μM), which detects TC by fluorescence quenching [11]. Fu et al. formulated a dual-response probe, CDs@ZIF-8/CdTe@MIP, showing an 11-fold reduction in the LOD compared to CD/CdTe@MIP due to the presence of ZIF-8, enabling an ultra-sensitive, selective, and visual detection of 4-nitrophenol (4-NP) [12]. These findings underscore the capability of ratiometric fluorescence sensors to exploit their inherent self-calibration features to effectively avoid interference from non-targeting factors, thereby greatly enhancing detection accuracy and reliability [13].

Typically, ratiometric fluorescence detection relies on a stable fluorescence signal and another signal that either enhances or diminishes in the presence of the target molecule or an isotropic signal response. Conversely, ratiometric fluorescent probes with reversed fluorescence intensity changes (where one emitting signal decreases while the other increases) greatly enhance detection sensitivity and accuracy owing to the intrinsic signal amplification effect [14]. Especially under UV light, the color change induced by such a probe is more visibly pronounced, facilitating easier identification and differentiation of the results [15].

Metal–organic frameworks (MOFs) have received increasing attention due to their high porosity, large specific surface area, tunable structure, and optical and thermal stability [11]. In recent years, Zr-based zirconium MOFs have been extensively utilized in heavy metals, drugs, and organic dye adsorption, owing to the Zr-O bond’s high stability against water, acid, alkali, and heat [16]. Nowadays, lanthanide metal ions (Ln^3+^) are commonly doped into Zr-MOF to improve their detection performance due to their excellent luminescent properties. Thomas Kasper et al. explored the effectiveness of rapid optical sensing of dioxygen by utilizing the Eu^3+^ synthetically modified Zr-MOF [17]. Wang et al. synthesized g-C_3_N_4_/UiO-66 (Zr/Ce) by employing Zr and Ce as metal sources for efficient photocatalytic CO_2_ reduction in visible light [18]. Nonetheless, compared to Eu^3+^ and Ce^3+^, the emission bands of Tb^3+^ show superior spectral separation, affording Tb^3+^ a unique advantage in fluorescent labeling, bio-imaging, and optical sensing [19]. In view of this, Zr^4+^ and Tb^3+^ can be intelligently and organically combined to develop a MOF material that inherits zirconium-based MOFs’ high stability and porosity while possessing Tb^3+^’s excellent luminescence performance and sensing function.

In this study, Tb^3+^ and Zr^4+^ were coordinated with 2-hydroxyterephthalic acid (BDC-OH) to synthesize UiO-OH@Tb for the ratiometric fluorescence detection of MG in fish samples (Figure 1). Upon interaction with MG, the fluorescence color of the probe transitioned from lime green to blue, owing to the FRET interaction between Tb^3+^ and MG, enhancing MG’s blue fluorescence while weakening Tb^3+^’s green fluorescence. This serves as a visual signal for the sensitive detection of MG. Meanwhile, the RGB value of the fluorescence image was analyzed by a smartphone, enabling the on-site rapid detection of MG concentrations in fish.

## 2. Materials and Methods

### 2.1. Chemicals and Materials

Zirconium tetrachloride (ZrCl_4_), terbium(III) chloride hexahydrate (TbCl_3_·6H_2_O), 2-hydroxyterephthalic acid (BDC-OH), 1,4-terephthalic acid (H_2_BDC), malachite green (MG), sulfadiazine (SD), sulfadimethoxine (SDM), sulfamethazine (SM2), sulfametoxydiazine (SMD), sulfapyridine (SPY), sulfamethoxazole (SMZ), sulfamethoxypyridazine (SMP), sulfaquinoxaline (SQX), doxycycline hyclate (DOX), penicillin (PEN), chloramphenicol (CAP), nitrofurazone (NF), furazolidone (FRZ), cyclosporin (CsA), chlortetracycline (CTE), kanamycin (KAN), and *N*,*N*-dimethylformamide (DMF) were purchased from Aladdin Industrial Co., Ltd. (Shanghai, China). All chemicals were of analytical reagent grade and used as received without further purification. Fresh crucian and perch were purchased from a local supermarket (Chongqing, China).

### 2.2. Apparatus and Measurements

The morphology and particle size distributions of UiO-OH and UiO-OH@Tb were meticulously analyzed using a scanning electron microscope (SEM, Hitachi Regulus 8100, Chiyoda City, Japan). Elemental analyses of the prepared UiO-OH@Tb were carried out using energy-dispersive spectroscopy (EDS, EDAX, Octane Elect Super-70 mm^2^, Philadelphia, PA, USA). X-ray photoelectron spectroscopy (XPS, ESCALAB 250Xi, Thermo Fisher Scientific, Waltham, MA, USA) further elucidated the chemical compositions and contents of UiO-OH and UiO-OH@Tb. The functional groups and chemical structures of UiO-OH and UiO-OH@Tb were evaluated by Fourier transform infrared spectroscopy (FTIR, Spectrum Two, PerkinEimer, Shelton, CT, USA), which provided a detailed understanding of the samples at the molecular level. The fluorescence lifetimes of both UiO-OH and UiO-OH@Tb, before and after the addition of MG, were precisely measured using a steady-state/transient fluorescence spectrometer (FLS980, Edinburgh Instruments, Livingston, UK), yielding crucial data for comprehending their photophysical behavior. Additionally, the fluorescence intensities of UiO-OH@Tb at varying concentrations were documented using a fluorescence spectrophotometer (F-7100, Hitachi High-Technologies, Minato City, Japan), further validating the photometric analysis.

### 2.3. Preparation of UiO-OH, UiO-OH@Tb, Tb-MOF, and Tb-OH

UiO-OH@Tb was synthesized according to a previously reported method with simple modifications [20]. ZrCl_4_ (93 mg), TbCl_3_·6H_2_O (74.5 mg), and BDC-OH (146 mg) were dissolved in 15 mL of DMF, and then 1 mL of concentrated hydrochloric acid was added under sonication until complete dissolution. Finally, the mixed solution was transferred to a 25 mL Teflon-lined autoclave and heated at 120 °C for 24 h. The solution was cooled to room temperature to obtain a milky white liquid. The obtained solution was centrifuged at 10,000 rpm for 10 min, and the precipitate was collected. The precipitates were then thoroughly washed with ethanol and DMF three times, respectively, and finally, the precipitates were collected. The product was dried in a vacuum oven at 60 °C for 12 h to obtain a white powdery product and stored at 4 °C away from light. UiO-OH was synthesized in parallel under the above conditions, except for the absence of TbCl_3_·6H_2_O. Tb-OH was synthesized in the absence of ZrCl_4_, and Tb-MOF was prepared by replacing the ligand BDC-OH with H_2_BDC, while ZrCl_4_ was not added. Apart from this, the other conditions for the synthesis of both MOFs were similar to those of UiO-OH@Tb.

### 2.4. Ratiometric Fluorescence Detection of MG with UiO-OH@Tb

The improved detection of MG was performed as described by Yue et al. [2]. The MG solution was initially prepared using deionized water as a solvent. A suspension of UiO-OH@Tb powder (5 mg) in 5 mL of ethanol was prepared under sonication to achieve a concentration of 1 mg/mL. Subsequently, 1000 μL of the freshly prepared UiO-OH@Tb suspension was mixed with a specific volume of MG (1 mM) to prepare different concentrations (2, 4, 6, 8, 10, 20, 40, 60, 80, 100, 150, and 200 μM) of the test solutions. The resulting mixture was thoroughly mixed for 10 min, and then the fluorescence intensity of the mixture at 450 nm and 548 nm was measured using an excitation wavelength of 335 nm.

### 2.5. Determination of MG in Real Fish Samples

To assess the effectiveness of the UiO-OH@Tb fluorescent probe, carp and perch were selected as representatives of freshwater and marine fish, respectively. The dorsal muscle of the fish was collected and homogenized in a blender. Then, 5 g of the homogenized sample was mixed with 25 mL of acetonitrile solution. The mixed solution was vortexed for 5 min, then sonicated for 15 min, and centrifuged at 10,000 rpm for 10 min. Finally, the supernatant was collected. Subsequently, 25 mL of hexane saturated with acetonitrile was thoroughly mixed with the supernatant to remove fat. The acetonitrile layer was selected for rotary evaporation, and the evaporated sample was dispersed and filtered through a 0.22 μM filter membrane for further analysis. The filtered fish sample solution was used to prepare MG at a 1 mM concentration. MG was added to achieve final concentrations of 5, 10, and 20 μM with a UiO-OH@Tb concentration of 1 mg/mL. Each sample was measured three times for each concentration under a reaction time of 30 s and a temperature of 25 °C. The mean value was calculated for further analysis. Finally, the spiked recoveries were calculated.

### 2.6. Detection of MG by Smartphone

A smartphone color detection application was used to measure the MG. First, 200 μL of UiO-OH@Tb (1 mg/mL) solution was added to the enzyme strip. Then, varying volumes of MG were added to the UiO-OH@Tb probe suspension, creating mixed solutions with concentrations ranging from 2 to 200 μM. After 30 s, the mixed solutions were exposed to UV light at 254 nm, and the resultant color was captured using a smartphone. The smartphone’s application converted the captured images into the red (R), green (G), and blue (B) color parameters. Finally, a linear equation correlated the blue/green ratio (B/G) to the concentration of MG.

### 2.7. Statistical Analyses

All experimental measurements were conducted with triplicate replication, and the outcomes are depicted as the mean value ± standard deviation. The graph plotting and the requisite statistical analyses were efficiently performed utilizing Origin 2021 software, leading to the formulation of valid and pertinent conclusions.

## 3. Results and Discussion

### 3.1. Characterizations of UiO-OH@Tb

The morphologies of the UiO-OH@Tb and UiO-OH were observed using scanning electron microscopy (SEM), revealing highly stacked particles (Figure 1A,B). The similar morphology of UiO-OH@Tb and UiO-OH indicates that Tb ion doping does not significantly alter UiO-OH’s structure. The EDS (Figure 1C) shows the presence of Zr, Tb, C, N, and O, confirming the successful synthesis of UiO-OH@Tb with a homogeneous distribution of Zr and Tb. The powder X-ray diffraction (PXRD) pattern of UiO-OH@Tb (Figure 1D), synthesized with Tb^3+^, is similar to that of UiO-OH, indicating that Tb^3+^ doping does not compromise the stability of zirconium-based MOFs [21].

In the Fourier transform infrared spectroscopy (FTIR) recordings (Figure 1E), the observed broad bands of the ligand BDC-OH in the range of 2550–2930 cm^−1^ are attributed to the O-H stretching vibrations in the carboxylate group [11]. Compared to BDC-OH, the broad hydroxyl peak of -COOH in UiO-OH and UiO-OH@Tb (2300–3700 cm^−1^) shows a blue shift and weakened intensity, suggesting that BDC-OH is partially deprotonated during the formation of MOFs [22]. The C=O stretching vibration of the carboxylate in BDC-OH (1668.8 cm^−1^) undergoes a significant red shift compared to UiO-OH (1659.09 cm^−1^), indicating the successful coordination of Zr^4+^ with the carboxylate group [23]. The C-O stretching vibrational peak of the phenolic hydroxyl group at 1247.17 cm^−1^ confirms its presence in the sample [20]. The negligible shift in UiO-OH@Tb compared to UiO-OH (1247.28 cm^−1^) suggests the successful replacement of some Zr^4+^ by Tb^3+^, coordinating with the -COOH group in BDC-OH.

Additionally, the surface elemental composition and chemical bonding of UiO-OH@Tb were analyzed using XPS. The XPS of UiO-OH@Tb (Figure 1F) exhibits characteristic Tb 3d peaks being absent in UiO-OH. The high-resolution XPS spectrum of C 1s (Appendix A) shows peaks at 284.75 eV (C-C/C=C), 286.46 eV (C=O), and 288.71 eV (C-O). The high-resolution XPS spectrum of N 1s (Appendix A) shows peaks at 400.62 eV (N-H) and 402.77 eV (N-O). The high-resolution XPS spectrum of O 1s (Appendix A) shows peaks at 533.89 eV (Zr-O) and 531.95 eV (C=O). Compared to pre-doping, the binding energy of the Zr-O bond in O 1s decreased from 534.08 (Appendix A) to 533.89 eV (Appendix A), and the binding energy of C=O increased from 531.81 to 531.95 eV, further demonstrating Tb^3+^ coordination with the carboxyl group of BDC-OH [1], which is consistent with the FTIR results. The Zr 3d spectrum of UiO-OH@Tb (Appendix A) shows peaks at 185.29 eV (Zr 3d_3/2_) and 182.92 eV (Zr 3d_5/2_). The Tb 3d spectrum (Appendix A) shows peaks at 1277.89 (Tb 3d_3/2_), 1250.81 (Tb 3d_5/2_), and 1243.23 eV (Tb 3d_5/2_), confirming the presence of Tb^3+^ in UiO-OH@Tb, which is consistent with the EDS results [24].

These observations strongly indicate successful Tb^3+^ doping in UiO-OH without altering its morphology or crystal structure. Tb^3+^ successfully replaced some Zr^4+^, coordinating with the -COOH group in the ligand, confirming the successful construction of UiO-OH@Tb.

### 3.2. Fluorescence Properties of UiO-OH@Tb

To thoroughly investigate UiO-OH@Tb’s ability to detect MG, the optical properties of the fluorescent probe were analyzed, and the experimental conditions were optimized beforehand. At room temperature, 1 mL of the UiO-OH@Tb solution (1 mg/mL) was placed in a cuvette to observe the fluorescence intensity at different excitation wavelengths. Appendix A illustrates the fluorescence spectra of UiO-OH@Tb at different excitation wavelengths, revealing an optimal excitation wavelength of 335 nm and corresponding emission positions of 450 nm, 492 nm, 548 nm, 588 nm, and 624 nm. Ethanol was determined to be the optimal solvent for UiO-OH@Tb, based on the fluorescence intensity ratio (F_450_/F_548_) of UiO-OH@Tb (1 mg/mL) in different buffer solutions at 25 °C and an excitation wavelength of 335 nm (Appendix A).

The emission spectra of UiO-OH@Tb and UiO-OH were recorded under the optimized conditions (Figure 2A), with the insets showing the fluorescence colors of UiO-OH@Tb (right) and UiO-OH (left) under UV lamp irradiation. Compared to UiO-OH, the fluorescence intensity of UiO-OH@Tb is reduced near 450 nm while showing stronger characteristic peaks of Tb^3+^ near 492, 548, 588, and 624 nm. Normally, Ln^3+^ does not emit strong fluorescence due to the forbidden 4f-4f electronic transition [25]. However, when bound to specific organic ligands, the radiant energy absorbed by the ligands can be transferred to the Ln^3+^ through intramolecular energy transfer, resulting in strong fluorescence, a phenomenon known as the “antenna effect (AE)” [26].

In the experiments, Tb-MOF and Tb-OH were prepared with the H_2_BDC and BDC-OH ligands, respectively. The quantum yield (QY) of Tb-OH (16.99%) was found to be much higher than that of Tb-MOF (0.36%) (Appendix A). Additionally, comparing the fluorescence intensities of the two, it is clear that Tb-OH has a higher fluorescence intensity (Appendix A). Therefore, BDC-OH was chosen as an antenna to switch on the fluorescence of Tb^3+^. In UiO-OH@Tb, the characteristic luminescence of Tb^3+^ was determined by ligand–metal energy transfer; BDC-OH transfers part of its energy to Tb^3+^ after absorbing UV light, resulting in the green fluorescence of Tb^3+^ (Figure 2B).

The stability of the fluorescent probe is crucial for the sensitivity of the sensor. The fluorescence intensity ratio (F_450_/F_548_) of UiO-OH@Tb under UV irradiation at different time points (0, 5, 10, 20, 30, and 60 min) remained largely stable (Appendix A). In addition, the fluorescence intensity ratio remained stable, even when stored at 4 °C under light protection for up to 14 days (Appendix A), indicating excellent photostability of the probe. These results lay a solid foundation for the next stage of research and application.

### 3.3. Detection of MG by UiO-OH@Tb

The changes in fluorescence intensity of UiO-OH@Tb under different MG concentrations are shown in Figure 3A. As the MG concentration increases, the fluorescence intensity at 548 nm gradually decreases, while the intensity at 450 nm increases significantly. Based on this phenomenon, the fluorescence intensity ratio of F_450_/F_548_ is an effective signal for assessing changes in the MG concentration. The intensity ratio of UiO-OH@Tb (F_450_/F_548_) reveals a good linear relationship with the MG concentration from 0 to 200 μM (Figure 3B), with a linear correlation coefficient R^2^ of 0.9937. The LOD is 0.19 μM (*n* = 11), calculated at 3σ/k. Additionally, analysis of the CIE chromaticity diagram (Figure 3C) further confirmed that the color gradually shifted from lime green to blue with an increasing MG concentration, which is consistent with the results under UV irradiation (Figure 3B inset).

In contrast, the fluorescence intensity of UiO-OH alone decreases gradually with the concentration of MG (Appendix A), showing only a single blue fluorescence quenching at 450 nm (inset of Appendix A). A good linear relationship between UiO-OH and the MG concentration is also observed (Appendix A), but the detection limit is 25 μM. Compared with UiO-OH, UiO-OH@Tb, as a ratiometric fluorescent probe, not only has a lower detection limit and higher sensitivity but also provides obvious visual detection, proving the advantages of bimetallic MOFs in fluorescence sensing applications.

Notably, the addition of MG rapidly (within 30 s) increases the fluorescence intensity ratio (F_450_/F_548_) of the UiO-OH@Tb probe, which stabilizes in about 10 min (Figure 3D). Under UV irradiation, the fluorescence color changes from lime green to blue (Figure 3D inset). Therefore, UiO-OH@Tb is a suitable fluorescence detection material for MG due to its fast response, excellent sensitivity, and selectivity.

### 3.4. Selectivity of UiO-OH@Tb to MG

To assess the feasibility of the UiO-OH@Tb probe for the detection of MG in real food samples, its selectivity and immunity to interference were investigated. The effects of various potential interfering substances (SD, SDM, SM2, SMD, SPY, SMZ, SMP, SQX, DOX, PEN, CAP, NF, FRZ, CsA, CTE, and KAN) and/or MG on the UiO-OH@Tb probe were examined using the fluorescence intensities ratio (F_450_/F_548_) as the detection indicator. The result illustrates that the fluorescent probe exhibited a significant fluorescence effect only for MG (Figure 3E), while responses to other antibiotics were negligible. Additionally, anti-interference experiments showed that the fluorescence intensity ratios of the probe increase in the presence of various interfering substances mixed with MG (Figure 3F). It can be concluded that UiO-OH@Tb has excellent selectivity and superior anti-interference ability, highlighting its potential for the detection of MG in real-sample analysis.

### 3.5. Sensing Mechanism of the Sensor

An interesting phenomenon was observed: using the UiO-OH probe to detect MG resulted in fluorescence quenching at 450 nm (Appendix A). However, with the UiO-OH@Tb probe, MG detection led to a significant enhancement in the blue fluorescence at 450 nm and a decrease in green fluorescence at 548 nm (Appendix A). MG induces significantly different effects in these two fluorescent probes depending on the presence of Tb^3+^, warranting an in-depth exploration of their mechanisms.

Several major quenching mechanisms in fluorescence analysis include the crystal structure collapse, inner filter effect (IFE), fluorescence resonance energy transfer (FRET), and photoinduced electron transfer (PET) [2,27,28]. The PXRD mapping (Figure 4A,A’) displays that the structural framework of MOFs retains its original structural features after MG treatment, suggesting that fluorescence quenching is unrelated to structural collapse [29] and related to molecular level interactions.

To investigate this phenomenon, the excitation and emission spectra of the UiO-OH@Tb and UiO-OH probes were comparatively analyzed with the UV absorption spectrum of MG. Figure 4B shows that the UV–visible spectrum of MG has a broad absorption range from 260–680 nm, with an obvious overlapping with the emission and excitation spectra of UiO-OH@Tb. This overlap may cause an energy transfer from UiO-OH@Tb to MG and inhibit the absorption of the excitation energy by Tb^3+^, resulting in the quenching of its characteristic emission peak [24,30]. Similarly, the excitation and emission spectra of UiO-OH overlapped with the UV absorption spectrum of MG (Figure 4B’), suggesting that both the green fluorescence quenching of UiO-OH@Tb and blue fluorescence quenching of UiO-OH could be due to FRET or IFE. Fluorescence lifetimes of donor molecules can distinguish between FRET and IFE [28]. The fluorescence lifetime of the MG-treated UiO-OH@Tb was reduced (Figure 4C), and it can be concluded that the fluorescence quenching of Tb^3+^ in UiO-OH@Tb was mainly due to FRET. Stern–Volmer curves (I_0_/I = 1 + K[Q]) are established for the MG-treated UiO-OH probes. It is obvious that there is a certain linear relationship between MG and the probes (Appendix A inset), and no obvious change in the fluorescence lifetimes of their probes before and after the addition of MG (Figure 4C’). Thus, it can be concluded that the decrease in UiO-OH fluorescence is caused by IFE.

Given MG’s weak fluorescence (Appendix A) and electron-withdrawing N^+^ group, it is speculated that the energy transfer via FRET, upon addition, significantly enhances MG’s blue fluorescence around 450 nm. To test this hypothesis, XPS and FTIR analyses were performed on the probes before and after the MG addition. The XPS analysis (Figure 5A) verifies that the Tb 3d peaks in UiO-OH@Tb red-shifted from 1277.89 and 1243.23 eV to 1278.44 and 1243.61 eV after the MG treatment. Simultaneously, the vibrational peaks of -COO- at 1380 cm^−1^ in the FTIR spectra were significantly reduced and blue-shifted to 1373 cm^−1^ after the MG treatment (Figure 5B). These spectral changes confirm the coordination between Tb^3+^ and MG [23], forming oxygen-bridged heteropoly nuclear complexes [31]. This interaction enhanced the fluorescence of MG (Figure 5C).

In summary, the response mechanism of UiO-OH@Tb to MG is mainly attributed to the FRET effect. Specifically, the FRET effect between MG and Tb^3+^ effectively quenches the fluorescence at 548 nm and, at the same time, this energy transfer process significantly enhances the blue fluorescence of MG around 450 nm.

### 3.6. Application of UiO-OH@Tb on Real Foods

To confirm the applicability and validity of the sensor probes in complex food samples, spiking recovery experiments were carried out on deep-sea fish (perch) and freshwater fish (crucian). According to Table 1, the recoveries of fish samples spiked with MG ranged from 95.91% to 109.80%, with relative standard deviations (RSD) as low as 1.17–3.21%. Additionally, the HPLC results further confirmed the UiO-OH@Tb probe’s reliability and accuracy. A comparison with the sensors in the existing literature revealed that the UiO-OH@Tb probe is superior or at least comparable to most of the reported sensors in terms of both sensitivity and detection range (Table 2). These results indicate that the developed sensor probes are not only theoretically advantageous but also effective in practical applications for the rapid and accurate quantitative detection of MG residues in different fish samples.

### 3.7. Smartphone Detection of MG

The ease of capturing fluorescent images on smartphones has established them as the preferred tool for on-site fluorescent color testing, particularly in antibiotic colorimetric testing [9]. Under UV irradiation at 254 nm, a distinct fluorescence color change from lime green to blue was observed in the ratio-sensing probe after adding different concentrations of MG (Figure 6). Consequently, the ratio of the green channel (G value) to the blue channel (B value) was selected to calculate the MG concentration. The RGB values of the images were extracted using a smartphone colorimetric application, and the RGB conversion results agreed with the CIE results (Figure 3C). The B/G intensity ratio exhibited a strong linear relationship with the MG concentration (0–200 μM, R^2^ = 0.9850) (Figure 6), with a detection limit of 0.94 μM. The R^2^ value is lower than that of the UiO-OH@Tb probe, possibly due to the color detector application’s lower sensitivity compared to the fluorescence spectrometer [35]. Nevertheless, the R^2^ value was sufficient to accurately measure the MG content. The results demonstrate that MG can be effectively monitored in real time using an intelligent color recognition system combined with UiO-OH@Tb.

## 4. Conclusions

In summary, a dual-emission fluorescent probe (UiO-OH@Tb) was successfully synthesized via a hydrothermal method and applied to the fluorescence detection of MG in fish samples. As the MG concentration increases, the green fluorescence of Tb^3+^ in the bimetallic probe UiO-OH@Tb is gradually quenched, while the blue fluorescence of MG is significantly enhanced. This transition causes the fluorescence color of the probe to shift from lime green to blue, providing an intuitive visual indication for MG detection. In contrast, the monometallic UiO-OH probe exhibits single blue light quenching during MG detection. Investigations into the response mechanisms of UiO-OH@Tb and UiO-OH to MG revealed that the FRET between Tb^3+^ and MG is the primary driving force behind the response of the UiO-OH@Tb probe to MG, while the IFE is the main cause of the blue light quenching in the UiO-OH probe.

Comparing the two probes, UiO-OH@Tb exhibited not only an intuitive fluorescence response but also a fast response time (30 s), excellent selectivity, high sensitivity, wide linear detection range, and extremely low detection limit (0.19 μM). The detection of MG in real fish samples verified the composite probe’s excellent utility and reliability, with recoveries ranging from 95.91% to 109.80%. The fluorescence color change was instantly captured by the smartphone sensing platform, revealing that the B/G intensity ratio exhibited an excellent linear relationship with the MG concentration, with a detection limit of 0.94 μM. Based on these results, it is reasonable to conclude that UiO-OH@Tb, as a ratiometric fluorescent probe, holds significant potential and application prospects for MG detection in other food samples.

## Data Availability

The original contributions presented in the study are included in the article and Appendix A, further inquiries can be directed to the corresponding author.

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
