# Peer review of "Enhanced Sensitivity and Accuracy of Tb3+-Functionalized Zirconium-Based Bimetallic MOF for Visual Detection of Malachite Green in Fish"

_foods, 2024, doi:10.3390/foods13172855_

Round 1

Reviewer 1 Report

Comments and Suggestions for Authors

In this manuscript, the authors have described the synthesis and study of a zirconium-based MOF functionalized with Tb³⁺ as a dual-emissive ratiometric fluorescent probe (UiO-OH@Tb) and its application for detecting the highly toxic Malachite Green (MG), which is banned in many countries in aquaculture. Moreover; the authors synthesized several other MOFs to prove the sensing mechanism and its advantages. The fluorescence color of the probe shifts from lime green to blue with increasing concentrations of MG, a change that occurs due to the FRET mechanism between Tb³⁺ and MG. Interestingly, this phenomenon induces a shift in the probe's fluorescence from lime green to blue while simultaneously increasing the blue fluorescence of MG. This probe shows a low detection limit excellent selectivity, sensitivity,, and superior anti-interference ability. The authors also successfully employed this probe to detect MG in deep-sea fish (Perch) and freshwater fish (crucian). I recommend this manuscript in the Foods journal.

Comments:

i) In lines 222 and 223, the UiO-OH@Tb 222 (left) and UiO-OH (right) should be written as UiO-OH@Tb 222 (right) and UiO-OH (left) according to Figure 2A and the results discussion.

Author Response

Comments 1:  [In lines 222 and 223, the UiO-OH@Tb 222 (left) and UiO-OH (right) should be written as UiO-OH@Tb 222 (right) and UiO-OH (left) according to Figure 2A and the results discussion.]

Response1:[Thank you for asking us this question. I apologise for this mistake, I have amended " the fluorescence colors of UiO-OH@Tb (left) and UiO-OH (right)" to " the fluorescence colors of UiO-OH@Tb (right) and UiO-OH (left)" in the manuscript and have highlighted it in red.]

Reviewer 2 Report

Comments and Suggestions for Authors

In their paper, Zhou et al. report the hydrothermal synthesis of a Zr4+-based MOF doped with Tb3+ (UiO-OH@Tb), and the application of this fluorescent probe for detection and quantitation of toxic malachite green (MG) residues. This is done by monitoring the ratiometric response of UiO-OH@Tb upon addition of MG, on the basis of a FRET mechanism occurring from Tb3+ (decreasing green emission) to MG (increasing blue emission). The performance of the probe in terms of sensitivity is compared to that of the parent MOF (UiO-OH) and of other monometallic MOFs, and found to be superior. The facile on-site detection of MG using the RGB function of a smartphone is also presented.

 I consider this work in Foods once the following observations are addressed by the authors.

Analysis of IR spectra:

Pg. 5, line 183: “The broad hydroxyl peak of -COOH in BDC-OH (2535-2994 cm-1) is absent in UiO-OH@Tb (Figure 1E), indicating complete deprotonation of BDC-OH in the complex”

 The broad peak at 2500–3400 cm-1 comprises stretching vibrations from both C-H bonds (below 3000 cm-1) and O-H bonds (above 3000 cm-1); for reference, please see the assignment of IR bands of the related compound 2,5-dihydroxyterephthalic acid [J. Mol. Struct. 2022, 1264, 133174, doi:10.1016/j.molstruc.2022.133174].

In Fig. 1D from the ref. 30 cited in the article, the disappearance of the O-H stretching band is clearly visible. In Fig. 1E in the present manuscript, however, this band shifts to larger wavenumbers (more so in the case of UiO-OH) and decreases in intensity with respect to the C=O stretching band. If complete deprotonation occurs, how can the presence of this band in the UiO-OH and UiO-OH@Tb samples be accounted for?

 Pg. 5, line 185: “The peak at 1659.36 cm-1 in UiO-OH@Tb is blue-shifted compared to BDC-OH, suggesting coordination between the deprotonated ligand and Zr4+ ion centre [22].”

This is incorrect. In IR spectroscopy, a blue shift of a bond frequency corresponds to a shift to higher wavenumbers, and is associated with a strengthening of the bond. Reference 22 quoted by the authors contains the same mistake. Ref. 30 is correct, using the term “red shift” and indicating that weakening of the C=O bond indicates coordination.

 Pg. 5, line 187: “The C=O stretching band of UiO-OH at 1659.09 cm-1 shifts to 1659.36 cm-1 after Tb3+ doping, indicating successful coordination of Tb3+ with BDC-OH.”

In my opinion, a 0.27 cm-1 red shift is quite small for making any assumption. In reference 20 cited by the authors, even differences of 1 cm-1 are considered negligible.

 Pg. 9, line 330: “the vibrational peaks of -COO- at 1380 cm-1 in FT-IR spectra were significantly reduced and blue-shifted to 1373 cm-1 after MG treatment (Figure 5B). These spectral changes confirm the coordination between Tb3+ and MG [30], forming oxygen-bridged heteropoly nuclear complexes” – here as well please correct “blue shift” by “red shift”

 Fluorescence analysis:

 Pg. 6, line 222: “insets showing the fluorescence colors of UiO-OH@Tb (left) and UiO-OH (right)” – Shouldn’t the “left” and “right” assignments be reversed?

 Fig. S6. For the Stern-Volmer analysis, were the data corrected to account for the emission of MG increasing in the same spectral range monitored for UiO-OH quenching (400–500 nm)?

Minor corrections:

 Pg. 3, line 91, Materials and methods: when listing the chemicals used, some are written with capital first letter, while others with lowercase letter; please unify (I suggest lower cases).

 Pg. 3, line 104: “energy spectroscopy” should be “energy dispersive spectroscopy”; line 112: for the time-resolved spectrofluorimeter, please correct the instrument and manufacturer names: “FLS980, Edinburgh Instruments”

 Pg. 4, line 160: please correct to “the resultant color was captured”

 Pg. 5, line 215: replace “demonstrates” by “illustrates”

 Pg. 7, caption of Fig. 3: “(Malachite Green)” can be removed from the caption (line 262); line 263: please replace “treat” by “treated”

 The term more commonly employed than “internal filtration effect” or “internal filtering effect” is “inner filter effect”.

Author Response

Comments 1:[Analysis of IR spectra: Pg. 5, line 183: “The broad hydroxyl peak of -COOH in BDC-OH (2535-2994 cm-1) is absent in UiO-OH@Tb (Figure 1E), indicating complete deprotonation of BDC-OH in the complex”

The broad peak at 2500–3400 cm-1 comprises stretching vibrations from both C-H bonds (below 3000 cm-1) and O-H bonds (above 3000 cm-1); for reference, please see the assignment of IR bands of the related compound 2,5-dihydroxyterephthalic acid [J. Mol. Struct. 2022, 1264, 133174, doi:10.1016/j.molstruc.2022.133174].

In Fig. 1D from the ref. 30 cited in the article, the disappearance of the O-H stretching band is clearly visible. In Fig. 1E in the present manuscript, however, this band shifts to larger wavenumbers (more so in the case of UiO-OH) and decreases in intensity with respect to the C=O stretching band. If complete deprotonation occurs, how can the presence of this band in the UiO-OH and UiO-OH@Tb samples be accounted for?

Pg. 5, line 185: “The peak at 1659.36 cm-1 in UiO-OH@Tb is blue-shifted compared to BDC-OH, suggesting coordination between the deprotonated ligand and Zr4+ ion centre [22].”

This is incorrect. In IR spectroscopy, a blue shift of a bond frequency corresponds to a shift to higher wavenumbers, and is associated with a strengthening of the bond. Reference 22 quoted by the authors contains the same mistake. Ref. 30 is correct, using the term “red shift” and indicating that weakening of the C=O bond indicates coordination.

Pg. 5, line 187: “The C=O stretching band of UiO-OH at 1659.09 cm-1 shifts to 1659.36 cm-1 after Tb3+ doping, indicating successful coordination of Tb3+ with BDC-OH.”

In my opinion, a 0.27 cm-1 red shift is quite small for making any assumption. In reference 20 cited by the authors, even differences of 1 cm-1 are considered negligible.

Pg. 9, line 330: “the vibrational peaks of -COO- at 1380 cm-1 in FT-IR spectra were significantly reduced and blue-shifted to 1373 cm-1 after MG treatment (Figure 5B). These spectral changes confirm the coordination between Tb3+ and MG [30], forming oxygen-bridged heteropoly nuclear complexes” – here as well please correct “blue shift” by “red shift”]

Response1: [Thank you for your advices. I'm apologise to say that these issues are indeed an oversight in my work.

  1. On page 5, line 183, amend "The broad hydroxyl peak of -COOH in BDC-OH (2535-2994 cm-1) is absent in UiO-OH@Tb (Figure 1E), indicating complete deprotonation of BDC-OH in the complex" was changed to "In the Fourier Transform Infrared Spectroscopy (FTIR) recordings (Figure. 1E), the observed broad bands of the ligand BDC-OH in the range of 2550-2930 cm-1 are attributed to the O-H stretching vibrations in the carboxylate group[11]. Compared to BDC-OH, the broad hydroxyl peak of -COOH in UiO-OH and UiO-OH@Tb (2300-3700 cm-1) shows blue shift and weakened intensity, suggesting that BDC-OH is partially deprotonated during the formation of MOFs[22]
  2. On page 5, line 185, replace "The peak at 1659.36 cm-1 in UiO-OH@Tb is blue-shifted compared to BDC-OH, suggesting coordination between the deprotonated ligand and Zr4+ ion centre [22]." Replace with " The C=O stretching vibration of the carboxylate in BDC-OH (1668.8 cm-1) undergoes a significant rew shift compared to UiO-OH (1659.09 cm-1), indicating the successful coordination of Zr4+ with the carboxylate group[23].
  3. On page 5, line 187, "The C=O stretching band of UiO-OH at 1659.09 cm-1 shifts to 1659.36 cm-1 after Tb3+ doping, indicating successful coordination of Tb3+ with BDC-OH." was deleted from the manuscript.
  4. The word "blue shift" has been changed to "red shift" on page 9, line 330, of the manuscript.]

Comments 2: [Fluorescence analysis: Pg. 6, line 222: “insets showing the fluorescence colors of UiO-OH@Tb (left) and UiO-OH (right)” – Shouldn’t the “left” and “right” assignments be reversed?

Fig. S6. For the Stern-Volmer analysis, were the data corrected to account for the emission of MG increasing in the same spectral range monitored for UiO-OH quenching (400–500 nm)?]

Response 2: [Thank you for your advices. I'm apologise to say that these issues are indeed an oversight in my work.

  1. I have amended " the fluorescence colors of UiO-OH@Tb (left) and UiO-OH (right)" to " the fluorescence colors of UiO-OH@Tb (right) and UiO-OH (left)" in the manuscript and have highlighted it in red.
  2. On page 9, line 321, replace "The fluorescence lifetime of MG-treated UiO-OH@Tb decreased (Figure 4C), while that of UiO-OH (Figure 4C') remained unchanged. Therefore, the fluorescence quenching of Tb3+ in the UiO-OH@Tb is mainly due to FRET, while the ligand fluorescence quenching of UiO OH is due to IFE." was changed to "The fluorescence lifetime of MG-treated UiO-OH@Tb was reduced (Figure. 4C), and it can be concluded that the fluorescence quenching of Tb3+ in UiO-OH@Tb was mainly due to FRET. Stern-Volmer curves (I0/I=1+K[Q]) are established for the MG-treated UiO-OH probes. It is obvious that there is a certain linear relationship between MG and the probes (Figure. S6A inset), and no obvious change in the fluorescence lifetimes of their probes before and after the addition of MG (Figure. 4C'). Thus it can be launched that the decrease in UiO-OH fluorescence is caused by IFE."]

Comments 3: [Minor corrections: Pg. 3, line 91, Materials and methods: when listing the chemicals used, some are written with capital first letter, while others with lowercase letter; please unify (I suggest lower cases).

Pg. 3, line 104: “energy spectroscopy” should be “energy dispersive spectroscopy”; line 112: for the time-resolved spectrofluorimeter, please correct the instrument and manufacturer names: “FLS980, Edinburgh Instruments”

Pg. 4, line 160: please correct to “the resultant color was captured”

Pg. 5, line 215: replace “demonstrates” by “illustrates”

Pg. 7, caption of Fig. 3: “(Malachite Green)” can be removed from the caption (line 262); line 263: please replace “treat” by “treated”

The term more commonly employed than “internal filtration effect” or “internal filtering effect” is “inner filter effect”.]

Response 3: [Thank you for your advices. We apologise for the above problems and have now made the corresponding changes to the manuscript as requested, and the changes are highlighted in red.

  1. On page 3, in line 91, the manuscript for chemical reagents have been changed to lowercase "terbium(III) chloride hexahydrate (TbCl3·6H2O), 2-hydroxyterephthalic acid (BDC-OH), 1,4-terephthalic acid (H2BDC), malachite green (MG), sulfadiazine (SD), sulfadimethoxine (SDM), sulfamethazine (SM2), sulfametoxydiazine (SMD), sulfapyridine (SPY), sulfamethoxazole (SMZ), sulfamethoxypyridazine (SMP), sulfaquinoxaline (SQX), doxycycline hyclate (DOX), penicillin (PEN), chloramphenicol (CAP), nitrofurazone (NF), furazolidone (FRZ)"; in line 104 under the same page number, "energy spectroscopy" has been changed to "energy dispersive spectroscopy"; the instrument and manufacturer of the steady state/transient fluorescence spectrometer in line 112 has also been corrected to read "FLS980, Edinburgh Instruments".
  2. On page 4, line 160, amend "the resultant colour were captured" to "the resultant color was captured".
  3. On page 5, line 215, "demonstrates" was replaced with "illustrates".
  4. On page 7, in the caption of Figure 3, "(Malachite Green)" was deleted and replaced "treat" with "treated" in line 263.
  5. On page 1, line 16, and on page 8, line 301, replaced "internal filtration effect"、" internal filtering effect " with "inner filter effect".]